# A high-throughput microsphere-based immunoassay of anti-SARS-CoV-2 IgM testing for COVID-19 diagnostics

**Dayu Zhang, Tianyang Xu, Eric Chu, Aiguo Zhang, Jinwei Du**[ORCID]*, **Michael Y. Sha**[ORCID]*

DiaCarta Inc, Richmond, CA, United States of America

* jdu@diacarta.com (JD); msha@diacarta.com (MYS)

## Abstract

The pandemic of novel coronavirus disease COVID-19 is rapidly expanding across the world. A positive result of antibody tests suggests that the individual has potentially been exposed to SARS-CoV-2, thus allowing to identify asymptomatic infections and determine the seroprevalence in a given population. The aim of this study was to evaluate the performances of a newly developed high throughput immunoassay for anti-SARS-CoV-2 IgM antibody detection on the Luminex MAGPIX platform. Clinical agreement studies were performed in 42 COVID-19 patient serum samples and 162 negative donor serum/plasma samples. Positive percent agreement (PPA) was 42.86% (95% CI: 9.90% to 81.59%), 71.43% (95% CI: 29.04% to 96.33%), and 28.57% (95% CI: 13.22% to 48.67%) for samples collected on 0–7 days, 8–14 days, and 2–8 weeks from symptom onset, respectively. Negative Percent Agreement (NPA) was 97.53% (95% CI: 93.80% to 99.32%). There was no cross-reactivity with the SARS-CoV-2 IgG antibody. Hemoglobin (200 mg/dL), bilirubin (2 mg/dL), triglyceride (250 mg/dL) and EDTA (10 mM) showed no significant interfering effect on this assay. In conclusion, an anti-SARS-CoV-2 IgM antibody assay with high sensitivity and specificity has been developed. With the high throughput, this assay will speed up the anti-SARS-CoV-2 IgM testing.

## Introduction

The SARS-CoV-2 virus has been identified as the cause of a respiratory illness outbreak in Wuhan, China in late 2019 and has since evolved into a global pandemic, COVID-19. As of early November 2020, 47 million people have contracted the virus and more than 1 million people have died. Nucleic acid amplification testing methods such as PCR have been the gold standard for COVID-19 detection during the early phase of infection. However, there is an increasing demand for antibody detection for determining the seroprevalence of COVID-19 in the general population. The shortage of swabs and nucleic acid detection kits in certain areas have also evoked the appreciation of serology tests. The SARS-CoV-2 serostatus of asymptomatic patients or patients with symptoms appearing late into the infection is of particular interest. The government and the media have been promoting positive antibody tests as an alternative or additional screening standard for individuals returning to work. A recent

**Data Availability Statement:** All relevant data are within the manuscript.

**Funding:** The funder (DiaCarta Inc) provided support in the form of salaries for authors [JD, DZ,

TX, EC, AZ and MS], but did not have any additional role in the study design, data collection and analysis, decision to publish, or preparation of the manuscript.

**Competing interests:** The funder (DiaCarta Inc) provided support in the form of salaries for authors [JD, DZ, TX, EC, AZ and MS], but did not have any additional role in the study design, data collection and analysis, decision to publish, or preparation of the manuscript. This does not alter our adherence to PLOS ONE policies on sharing data and materials.

study demonstrated that both IgM and IgG antibodies were detectable 5 days after onset in all 39 patients with SARS-CoV-2 infection [1], and the median day of seroconversion for both IgG and IgM was 13 days post symptom onset [2]. The presence of IgM antibodies can indicate an active or recent infection while the presence of IgG antibodies usually signals past infection. Ultimately, serological testing can help detect cases of SARS-CoV-2 for which PCR testing resulted in false negatives, identify asymptomatic infections, confirm results for clinically suspicious cases, and help guide return-to-work or travelling decisions [3].

Herein, we reported the performance evaluation of the QuantiVirus™ anti-SARS-CoV-2 IgM test which is a two-step immunoassay using Luminex platform to detect anti-SARS-CoV-2 spike protein 1 (S1) receptor-binding domain (RBD) IgM antibody in human serum or plasma specimens. Validation of the test was conducted using COVID-19 negative and positive samples on MAGPIX® instruments. The test takes approximately 3 hours per run with a 96-well plate capable of testing 92 patient samples, enabling a streamlined workflow for high-throughput COVID-19 antibody testing.

## Methods and materials

### Instrumentation

According to the guidance issued by Centers for Disease Control (CDC) and the World Health Organization (WHO), all studies were conducted in a Biosafety Level 2 (BSL-2) cabinet when handling COVID-19 patient samples. The microplate shaker (PlexBio Co, Taiwan) was used for microplate shaking and incubation. Data acquisition was performed on Luminex MAG-PIX® instruments (Luminex, Austin, TX).

### Reagents and patient samples

The recombinant SARS-CoV-2 Spike protein 1 (RBD)-His was produced from HEK293 suspension cells (Innovative Research, Inc, MI). Anti-SARS-CoV-2 Spike RBD monoclonal antibody (IgM isotype) was purchased from Creative Diagnostics (Shirley, NY). PE conjugated anti-human IgM antibody was purchased from BioLegend (San Diego, CA). MagPlex Microsphere and xMAP® Antibody Coupling (AbC) kit was purchased from Luminex (Austin, TX). Hemoglobin (human), bilirubin, triglyceride and EDTA were purchased from Sigma-Aldrich (St. Louis, MO). BlockAid™ Blocking Solution was purchased from ThermoFisher Scientific (Waltham, MA). 96-well microplates (flat bottom, clear) were purchased from Greiner bio-one (Monroe, NC).

Healthy donor EDTA K2 plasma samples were purchased from Golden West Biosolutions (Temecula, CA) and healthy donor serum samples were purchased from Innovative Research, LLC (Plymouth, MN). COVID-19 patient serum samples were confirmed by DIAZYME SARS-CoV-2 IgG/IgM CLIA kit and acquired from ProMedDx (Norton, MA).

### Assay procedure

Principle of the assay is shown in Fig 1. Recombinant spike protein 1 (S1) RBD was covalently coupled to the surface of MagPlex® Microspheres (magnetic beads) via a carbodiimide linkage using xMAP® Antibody Coupling (AbC) kit. First, 3 μL of S1 RBD protein coated magnetic beads, 87 μL of BlockAid™ Blocking Solution and 10 μL of serum or plasma samples were loaded to 96-well plate and incubated at room temperature for 1 hour with shaking at 600rpm. The IgM antibodies present in human specimens against S1 RBD protein (the antigen) will bind to the coated magnetic beads. After washing, PE conjugated anti-human IgM antibody was added to the reaction mixture and incubated at room temperature for 30 minutes with

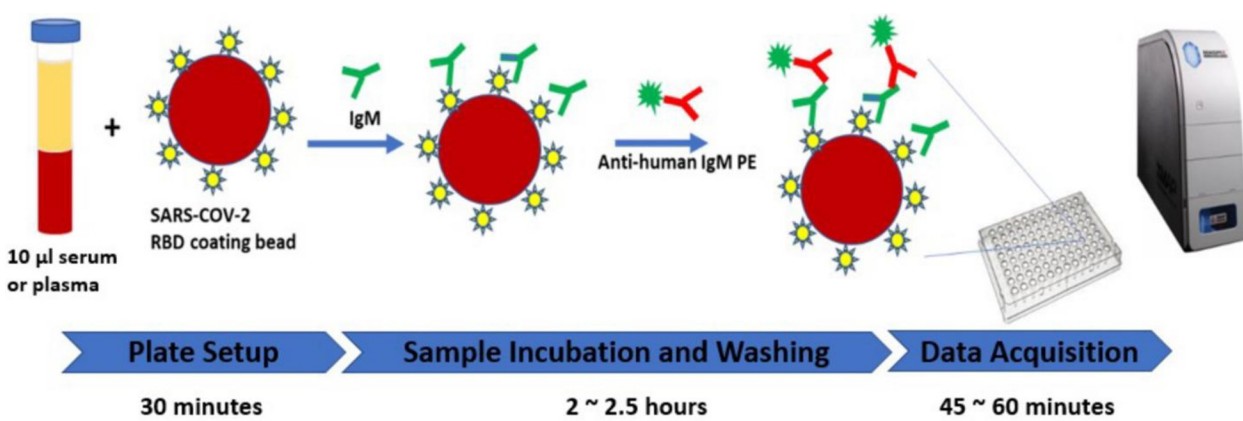

**Fig 1. The high throughput immunoassay for anti-SARS-CoV-2 IgM detection.**

shaking at 600rpm. After washing, data acquisition was performed on MAGPIX® instrument for Median Fluorescence Intensity (MFI). Interpretation of the testing results was performed by calculating the ratio between the MFI of each sample and the average MFI of two blank wells loaded with 10 μL PBS-1% BSA instead of serum or plasma samples. The cut-off value was calculated based on the ratio distribution of 162 COVID-19 negative samples. The sample will be interpreted as positive when the ratio is $\geq$45; otherwise, the sample will be considered negative for Anti-SARS-CoV-2 IgM antibody.

## Performance evaluation

To evaluate the clinical performance of the QuantiVirus$^{TM}$ Anti-SARS-CoV-2 IgM Test, 162 COVID-19 negative samples and 42 COVID-19 positive samples were tested and evaluated for NPA and PPA.

Class specificity was evaluated by testing ten IgG-positive and IgM-negative patient samples. Within-run precision (repeatability) was evaluated by testing negative sample and positive sample in 20 replicates. Between-run precision was evaluated by testing negative sample and positive sample on five separate runs with two replicates per run. For interference testing, hemoglobin (200 mg/dL), bilirubin (2 mg/dL), triglyceride (250 mg/dL) and EDTA (10mM) were spiked into serum samples, respectively, and the MFI was compared with the control samples.

## Statistical analysis

For precision evaluations, coefficient of variation (CV) was calculated as the ratio of the standard deviation (SD) to the mean. For interference testing, the samples spiked with potential interfering substances were compared with the control samples by paired T-test with $p \leq 0.05$ defined as significantly different.

## Results

### Clinical performance

Forty-two (42) serum samples collected at different times from individuals who tested positive with a RT-PCR method for SARS-CoV-2 infection were used in the evaluation of positive percent agreement (PPA). One hundred and sixty-two (162) serum or EDTA plasma samples collected from healthy donors were used to establish the cut-off between positive and negative

**Table 1. Positive Percent Agreement (PPA) and Negative Percent Agreement (NPA).**

| Category | Days from Symptom Onset | Number of Samples | IgM Positive | IgM Negative | PPA and NPA (95% CI) |
|---|---|---|---|---|---|
| COVID-19 Positive | 0–7 days | 7 | 3 | 4 | PPA: 42.86% (9.90% to 81.59%) |
| | 8–14 days | 7 | 5 | 2 | PPA: 71.43% (29.04% to 96.33%) |
| | 2–8 weeks | 28 | 8 | 20 | PPA: 28.57% (13.22% to 48.67%) |
| COVID-19 Negative | n/a | 162 | 4 | 158 | NPA: 97.53% (93.80% to 99.32%) |

n/a: Not applicable.

results and were in the evaluation of Negative Percent Agreement (NPA). As shown in Table 1, PPA was 42.86% (95% CI: 9.90% to 81.59%), 71.43% (95% CI: 29.04% to 96.33%), and 28.57% (95% CI: 13.22% to 48.67%) for samples collected on 0–7 days, 8–14 days, and 2–8 weeks from symptom onset, respectively, and NPA was 97.53% (95% CI: 93.80% to 99.32%). The Area under the ROC Curves (AUCs) are 0.85, 0.67, and 0.38 for samples collected on 0–7 days, 8–14 days, and 2–8 weeks from symptom onset, respectively (Fig 2).

In addition, 4 pairs of matched serum and EDTA plasma samples (i.e. collected from the same COVID-19 patients) were tested with QuantiVirus[TM] Anti-SARS-CoV-2 IgM Test and 100% concordance was observed, as shown in Table 2. It indicates that EDTA plasma is as acceptable as serum for this test.

## Interfering substance

Hemoglobin (200 mg/dL) was spiked into three serum samples to test the potential interfering effect of high-level hemoglobin which might be present in hemolysis and other conditions. Bilirubin (2 mg/dL) was spiked into three serum samples to test the potential interfering effect of

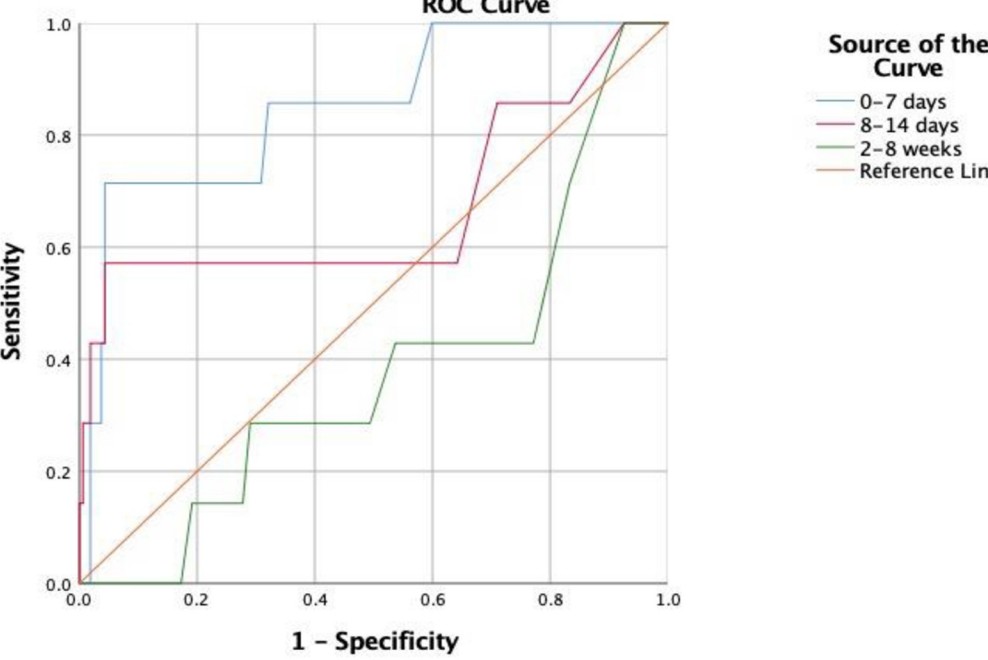

**Fig 2. Receiver operating characteristic (ROC) curve and area under the curve (AUC) to evaluation the performance of the assay.**

**Table 2. Comparison between 4 pairs of matched serum and EDTA plasma samples.**

| Sample ID | Serum Sample | | EDTA Plasma Sample | |
|---|---|---|---|---|
| | Ratio to Blank | Interpretation | Ratio to Blank | Interpretation |
| ProMed#p62 | 7 | Negative | 5 | Negative |
| ProMed#p63 | 111 | Positive | 107 | Positive |
| ProMed#p64 | 63 | Positive | 130 | Positive |
| ProMed#p65 | 65 | Positive | 120 | Positive |

high-level bilirubin in the blood which might be caused by liver dysfunction such as hepatitis and cirrhosis. Triglyceride (250 mg/dL) was spiked into three serum samples to test the potential interfering effect of high triglyceride levels which are often a sign of conditions that increase the risk of heart disease and stroke, including obesity and metabolic syndrome. Lastly, EDTA (10mM) was spiked into three serum samples to test the potential interfering effect of EDTA which is the anticoagulant used in EDTA blood collection tubes.

As shown in Table 3, the difference in the fluorescence signal (MFI) between the control samples and the samples spiked with hemoglobin, bilirubin, triglyceride or EDTA was all $\leq 15.0\%$ which is acceptable to the test and no false negative or false positive results were observed.

## Precision

Intra-assay reproducibility (repeatability) was evaluated by testing negative sample and positive sample in 20 replicates and the CV% of MFI was calculated. Inter-assay reproducibility (between-run precision) was evaluated by testing one negative sample and two positive

**Table 3. Interfering substance testing.**

| Sample | Group | MFI-1 | MFI-2 | Avg | Difference between control and tested substance |
|---|---|---|---|---|---|
| Positive Sample #1 | Control | 888 | 837 | 863 | |
| | EDTA | 875 | 841 | 858 | -0.6% |
| | Hemoglobin | 722 | 800 | 761 | -11.8% |
| Positive Sample #2 | Control | 1340 | 1378 | 1359 | |
| | EDTA | 1318 | 1276 | 1297 | -4.6% |
| | Hemoglobin | 1220 | 1155 | 1188 | -12.6% |
| Negative sample | Control | 64 | 68 | 66 | |
| | EDTA | 59 | 54 | 57 | -13.6% |
| | Hemoglobin | 55 | 59 | 57 | -13.6% |
| Positive Sample #1 | Control | 793 | 769 | 781 | |
| | Bilirubin | 779 | 867 | 823 | 5.4% |
| Positive Sample #2 | Control | 1041 | 1240 | 1140 | |
| | Bilirubin | 985 | 1186 | 1085 | -4.8% |
| Negative sample | Control | 47 | 48 | 47 | |
| | Bilirubin | 45 | 49 | 47 | -0.5% |
| Positive Sample #1 | Control | 682 | 696 | 689 | |
| | Triglyceride | 670 | 745 | 708 | 2.7% |
| Positive Sample #2 | Control | 1209 | 1099 | 1154 | |
| | Triglyceride | 1166 | 1107 | 1137 | -1.5% |
| Negative sample | Control | 54 | 49 | 52 | |
| | Triglyceride | 47 | 47 | 47 | -8.7% |

**Table 4. Intra-assay reproducibility.**

| Sample | MFI of each of 20 replicates | | | | | | | | | | Avg | SD | CV% |
|--------|------|------|------|------|------|------|------|------|------|------|------|-----|------|
| Serum#36 | 1717 | 1147 | 1210 | 1404 | 1391 | 1263 | 1361 | 1655 | 1674 | 1494 | 1394 | 189 | 13.5% |
|  | 1412 | 1285 | 1348 | 1550 | 1500 | 1437 | 1618 | 1196 | 1077 | 1146 | | | |
| Serum#32 | 985 | 1295 | 988 | 1020 | 1172 | 1045 | 1060 | 973 | 721 | 887 | 1024 | 148 | 14.4% |
|  | 929 | 769 | 869 | 1170 | 1083 | 1212 | 1198 | 1093 | 1072 | 943 | | | |
| Serum_N | 40 | 43 | 41 | 41 | 42 | 40 | 46 | 42 | 41 | 44 | 42 | 3 | 6.4% |
|  | 39 | 43 | 43 | 41 | 44 | 39 | 35 | 38 | 43 | 46 | | | |

samples on five separate runs and the CV% of the Ratio of MFI $_{Sample}$ to MFI $_{Blank}$ was calculated. As shown in Tables 4 and 5, the CV% of within-run precision and between-run precision was below 15.0% and 20.0%, respectively.

## Class specificity

Antibody class specificity between IgM and IgG was tested for cross reactivity. Ten SARS--CoV-2 patient samples which were positive for IgG and negative for IgM (tested by DIAZYME SARS-CoV-2 IgG and IgM CLIA kit) were tested with QuantiVirus™ anti-SARS-CoV-2 IgM test and all were negative, thus establishing the specificity of the QuantiVirus™ anti-SARS--CoV-2 IgM test to the IgM class of antibodies (Table 6).

## Discussion

Antibody tests are blood tests that detect antibodies or immunoglobulins that are produced as the human immune response to SARS-CoV-2 infection. Antibody testing has multiple essential roles: it can identify asymptomatic infections, verify that vaccines are working, be used in contact tracing after a suspected infection in an individual, and to help inform public policy makers how many asymptomatic cases have occurred in a population [4–6]. As of October 28, 2020, more than 50 serological tests have been approved by FDA for emergency use authorization (EUA) for the detection of anti-SARS-CoV-2 IgM, IgG, or total antibodies and in various formats including lateral flow immunoassays (LFAs), enzyme-linked immunosorbent assays (ELISAs), chemiluminescent immunoassays (CLIA), and beads-based fluorescent immunoassays [7].

Despite the enormous efforts put by companies and researchers into developing serological assays, diagnostic accuracy of serological tests for COVID-19 has been largely variable. Bastos et al performed a systematic review and meta-analysis of published data on Medline, bioRxiv, and medRxiv from 1 January to 30 April 2020, and found that the pooled sensitivity of ELISAs measuring IgG or IgM was 84.3% (95% confidence interval 75.6% to 90.9%), of LFIAs was 66.0% (49.3% to 79.3%), and of CLIAs was 97.8% (46.2% to 100%) [8]. Furthermore, sensitivity was higher at least three weeks after symptom onset (ranging from 69.9% to 98.9%) compared

**Table 5. Inter-assay reproducibility.**

| Sample | Ratio of MFI $_{Sample}$ to MFI $_{Blank}$ | | | | | | | |
|--------|--------|--------|--------|--------|--------|------|------|------|
|  | Test-1 | Test-2 | Test-3 | Test-4 | Test-5 | Avg | SD | CV% |
| Serum#36 | 99.7 | 106.5 | 84.9 | 74.0 | 98.3 | 92.7 | 13.03 | 14.1% |
| Serum#32 | 72.9 | 48.1 | 53.9 | 46.8 | 53.6 | 55.1 | 10.47 | 19.0% |
| Serum_N | 3.0 | 3.6 | 4.1 | 3.2 | 3.1 | 3.4 | 0.48 | 14.1% |

**Table 6. Class-specificity test.**

| Sample | DIAZYME SARS-CoV-2 IgG/IgM CLIA kit | | QuantiVirus™ anti-SARS-CoV-2 IgM test | | |
|---|---|---|---|---|---|
| | SARS-CoV-2 IgG | SARS-CoV-2 IgM | MFI | Ratio of MFI $_{Sample}$ to MFI $_{Blank}$ | Interpretation |
| ProMed#p21 | positive | < 1 | 67 | 4 | negative |
| ProMed#p22 | positive | < 1 | 189 | 11 | negative |
| ProMed#p23 | positive | < 1 | 85 | 5 | negative |
| ProMed#p24 | positive | < 1 | 86 | 5 | negative |
| ProMed#p25 | positive | < 1 | 333 | 20 | negative |
| ProMed#p26 | positive | < 1 | 445 | 26 | negative |
| ProMed#p27 | positive | < 1 | 228 | 13 | negative |
| ProMed#p28 | positive | < 1 | 154 | 9 | negative |
| ProMed#p31 | positive | < 1 | 185 | 11 | negative |
| ProMed#p32 | positive | < 1 | 316 | 19 | negative |

with within the first week (from 13.4% to 50.3%). Similar findings have been confirmed by other investigators as well [9,10].

Of note, multiple publications have indicated that the appearance of detectable anti-SARS-CoV-2 IgM antibodies after infection with COVID-19 is delayed, resulting in abnormal sensitivity in the early days after the onset of symptoms. For instance, a group from Germany observed that less than 50% of patients produced detectable anti-SARS-COV-2 IgM in the first 10 to 14 days after the "onset" of symptoms [11]. Similarly, Long et al showed that only 12% to 40% of patients developed anti-SARS-COV-2 IgM seroconversion during 1 to 7 days post onset of symptom [12]. Using an ELISA designed to detect anti-SARS-COV-2 IgM antibodies against the RBD of the S1 subunit, data from Zhao et al. showed that approximately 28% of patients were IgM positive by day 7 whereas 73% turned positive by day 14 post symptom onset [13]. In agreement with those previous findings, using the QuantiVirus™ anti- SARS-CoV-2 IgM Test developed in our lab, we found that 42.86% of patients produced detectable IgM antibody in the first 7 days post symptom onset and increased to 71.43% by day 14. The test indicated a leading accuracy of report for the patient samples in the first 7 days, and a reliable report rate as the time extended.

To further establish the assay accuracy, and consistency against various patient conditions, we also showed that this microsphere -based fluorescence immunoassay has a high specificity of 97.53% and is compatible with both serum and EDTA-plasma samples. The interference of substances, such as hemoglobin, bilirubin, and triglyceride, would not generate inaccurate results. This indicates the potential application of our test on patients with certain health conditions.

QuantiVirus™ anti- SARS-CoV-2 IgM Test also demonstrated an excellent reproducibility, where the intra-assay variations are lower than 15% for positive samples, and 6.4% for the negative serum sample. The variations are below 20% when it comes to the comparison of readings among 5 different test runs. Therefore, QuantiVirus™ anti- SARS-CoV-2 IgM Test provides reliable, reproducible results for patient screening, confirmation, and tracing.

In addition, QuantiVirus™ anti- SARS-CoV-2 IgM Test provides a simplified workflow for clinical practices (Fig 1). The pre-coupled capture beads and reporting complex reduced the time-consuming operations of RBD binding. Reducing the workflow could further reduce the operational error introduced from RBD protein conjugation. The workflow can be quickly adapted by clinical practices to improve the throughput and can be easily integrated into a laboratory's clinical operations.

In conclusion, we have successfully developed a reliable high-throughput microsphere immunoassay for qualitative detection of anti-SARS-CoV-2 IgM antibody. The assay was

validated with COVID19 positive samples as well as negative samples obtained from healthy donors on MAGPIX® instrument. We believe that this assay will help to determine the infection status of COVID-19 and the true scope of the pandemic.

## Author Contributions

**Conceptualization:** Jinwei Du, Michael Y. Sha.

**Data curation:** Dayu Zhang, Tianyang Xu, Eric Chu.

**Formal analysis:** Dayu Zhang, Tianyang Xu, Eric Chu.

**Funding acquisition:** Aiguo Zhang.

**Investigation:** Michael Y. Sha.

**Methodology:** Jinwei Du, Michael Y. Sha.

**Project administration:** Aiguo Zhang.

**Resources:** Aiguo Zhang.

**Supervision:** Michael Y. Sha.

**Writing – original draft:** Dayu Zhang, Jinwei Du, Michael Y. Sha.

**Writing – review & editing:** Jinwei Du, Michael Y. Sha.

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
