## [Decision Letter · Decision Letter 0]

10 May 2021

PONE-D-21-09355

A High-throughput Microsphere-based Immunoassay of Anti-SARS-CoV-2 IgM Testing for COVID-19 Diagnostics

PLOS ONE

Dear Dr. Sha,

Thank you for submitting your manuscript to PLOS ONE. After careful consideration, we feel that it has merit but does not fully meet PLOS ONE’s publication criteria as it currently stands. Therefore, we invite you to submit a revised version of the manuscript that addresses the points raised during the review process.

We look forward to receiving your revised manuscript.

Kind regards,

Chandrabose Selvaraj, Ph.D.

Academic Editor

PLOS ONE

Journal Requirements:

We note that you have included the phrase “data not shown” in your manuscript. Unfortunately, this does not meet our data sharing requirements. PLOS does not permit references to inaccessible data. We require that authors provide all relevant data within the paper, Supporting Information files, or in an acceptable, public repository. Please add a citation to support this phrase or upload the data that corresponds with these findings to a stable repository (such as Figshare or Dryad) and provide and URLs, DOIs, or accession numbers that may be used to access these data. Or, if the data are not a core part of the research being presented in your study, we ask that you remove the phrase that refers to these data.

Thank you for stating the following in the Competing Interests section:

The authors have declared that no competing interests exist

We note that one or more of the authors are employed by a commercial company: DiaCarta Inc

Additional Editor Comments:

Now I have received the reviewer’s review reports and all three reviewers have recommended the manuscript for a substantial revision. Please review the manuscript in accordance with the reviewer's comments and submit, along with reference checks.

Reviewers' comments:

Reviewer's Responses to Questions

**Comments to the Author**

1. Is the manuscript technically sound, and do the data support the conclusions?

Reviewer #1: Partly

Reviewer #2: Yes

Reviewer #3: Partly

2. Has the statistical analysis been performed appropriately and rigorously? 

Reviewer #1: Yes

Reviewer #2: Yes

Reviewer #3: Yes

3. Have the authors made all data underlying the findings in their manuscript fully available?

Reviewer #1: No

Reviewer #2: Yes

Reviewer #3: No

4. Is the manuscript presented in an intelligible fashion and written in standard English?

Reviewer #1: Yes

Reviewer #2: Yes

Reviewer #3: Yes

5. Review Comments to the Author

Reviewer #1: This article is well written.

There are some missing informations and discussion.

Here are my suggestions:

2.3 Assay procedure

What are the two blank wells used to calculate the ratio ? You should describe if it is a human sample, or a synthetic sample.

Cut-off value is set to 45. You should indicate how this cut-off has been determined (same comment in the clinical performance section).

3.1 Clinical Performance

Table 1: please replace “IgG Positive” / “IgG Negative” by “IgM Positive” / ”IgM Negative”

Healthy donors data should be presented and discussed.

Data ROC curve and AUC analysis should be performed. The methodology used to determine the Cut-off value should be presented.

Instead of “data not shown”, data generated with matched serum and EDTA plasma should presented to demonstrate that EDTA plasma can be used.

3.3 Precision

“Inter-assay reproducibility (between-run precision) was evaluated by testing negative sample and positive sample on five separate runs “

Please correct: “Inter-assay reproducibility (between-run precision) was evaluated by testing one negative sample and two positive samples on five separate runs “

You should indicate in table 4 that ratio are presented. It would be interesting to give additional characterization of serum#36 and Serum#32 (for instance the level positivity on a reference test).

I would suggest to mention the stability of the developed reagents.

4. Discussion

You should address a major question:

Why do you use Luminex technology in a simplex mode ?

You might want to present some preliminary data of an IgM and IgG kit.

Reviewer #2: In this study, a high-throughput immunoassay of anti-SARS-CoV-2 IgM antibody was carried out on Luminex Magpix platform. A total of 42 serum samples from patients with positive COVID-19 were collected. This study has certain value for the control of SARS coronavirus. However, the authors should declare the creativity of this work and compare it in depth with other standard IgM tests。

Reviewer #3: The anti-SARS CoV2 IgM assay described is rather straight forward one. The investigators have used commercially available recombinant SARS CoV2 Spike protein 1 (RBD) and conjugated with commercially available microsphere and all reagents used in the assay were commercial ones. Some of the points that need clarification are:

1. What was the source of Covid positive and covid negative patient sample? How as Covid positivity or negativity determined? This information is crucial since all conclusions on performance are based on these samples. Were the patients RT-PCR positive or seropositive? If latter, what was the methodology used and the status of seropositivity (igM or IgG or both?).

2. As per section 3.4, there was data on igG positivity and IgM negativity for 10 samples from another commercial kit. What about others (totally 42 SARS CoV2 positive used).

3. The study was designed to test for SARS CoV2 IgM, but Table 1 shows data for IgG. What about IgM?

---

## [Author Response · Author response to Decision Letter 0]

25 Jun 2021

5. Review Comments to the Author

Reviewer #1: This article is well written.

There are some missing informations and discussion.

Here are my suggestions:

2.3 Assay procedure

What are the two blank wells used to calculate the ratio? You should describe if it is a human sample, or a synthetic sample.

The blank wells consist of 3 µL of S1 RBD protein coated magnetic beads, 87 µL of BlockAid™ Blocking Solution, and 10 µL PBS-1% BSA. This information was added to the manuscript’s Methods 2.3 Assay procedure.

.

Cut-off value is set to 45. You should indicate how this cut-off has been determined (same comment in the clinical performance section).

The cut-off value was calculated based the ratio distribution of 162 COVID-19 negative samples tested. This information was added to the manuscript ‘s Methods 2.3 Assay procedure.

. 

3.1 Clinical Performance

Table 1: please replace “IgG Positive” / “IgG Negative” by “IgM Positive” / ”IgM Negative”

Have replaced. Thank you.

Healthy donors data should be presented and discussed.

Data ROC curve and AUC analysis should be performed. The methodology used to determine the Cut-off value should be presented.

According to your suggestion, the ROC curve is drafted and shown in Figure 2. The Area under the ROC Curves (AUCs) are 0.85, 0.67, and 0.38 for samples collected on 0-7 days, 8-14 days, and 2-8 weeks from symptom onset, respectively. This information was added to the manuscript.

Instead of “data not shown”, data generated with matched serum and EDTA plasma should presented to demonstrate that EDTA plasma can be used.

According to your suggestion, the data is now presented in Table 2.

3.3 Precision

“Inter-assay reproducibility (between-run precision) was evaluated by testing negative sample and positive sample on five separate runs “

Please correct: “Inter-assay reproducibility (between-run precision) was evaluated by testing one negative sample and two positive samples on five separate runs “

Have corrected. Thank you.

You should indicate in table 4 that ratio are presented. It would be interesting to give additional characterization of serum#36 and Serum#32 (for instance the level positivity on a reference test).

Table 4 (now Table 5) now indicated that ratios are presented. 

I would suggest to mention the stability of the developed reagents.

we have tested the reagent stability and confirmed the shelf life of the kit is about 12 months.

4. Discussion

You should address a major question:

Why do you use Luminex technology in a simplex mode ?

You might want to present some preliminary data of an IgM and IgG kit.

Great question. We do have another kit for anti-SARS-CoV-2 IgG detection. Luminex Magpix is used in a simplex mode due to specific customer needs.

As a positive IgG test indicates protection against infection (i.e., sought after by people who are vaccinated) and a positive IgM test indicates recent infection (i.e., required by travelers flying to China), these two tests were rarely purchased together by our customers. 

Reviewer #2: In this study, a high-throughput immunoassay of anti-SARS-CoV-2 IgM antibody was carried out on Luminex Magpix platform. A total of 42 serum samples from patients with positive COVID-19 were collected. This study has certain value for the control of SARS coronavirus. However, the authors should declare the creativity of this work and compare it in depth with other standard IgM tests。

This is a microsphere-based immunoassay. It is semi-quantitative assay and can give a quantitative IgM (MFI) for each patient. This is particle important when we discuss this with vaccinated individual. We have demonstrated infected individual has same amount IgM for N protein and S1 protein. However, for recovered and vaccinated individual, anti SARS-CoV-2 N IgM and S1 IgM has different amount which it can be distinguished each other (our different manuscript). It can run 96 samples each time during 3 hrs. It is quite different to LFI in the throughput and accuracy. 

Reviewer #3: The anti-SARS CoV2 IgM assay described is rather straight forward one. The investigators have used commercially available recombinant SARS CoV2 Spike protein 1 (RBD) and conjugated with commercially available microsphere and all reagents used in the assay were commercial ones. Some of the points that need clarification are:

1. What was the source of Covid positive and covid negative patient sample? How as Covid positivity or negativity determined? This information is crucial since all conclusions on performance are based on these samples. Were the patients RT-PCR positive or seropositive? If latter, what was the methodology used and the status of seropositivity (igM or IgG or both?).

These samples were tested by FDA EUA IgM assay DIAZYME SARS-CoV-2 IgG/IgM CLIA kit and the patient were also confirmed by FDA EUA qPCR. We noted in section 2.2 Reagents and patient samples that COVID-19 patient serum samples were acquired from ProMedDx. We noted in section 3.1 Clinical Performance that the patients are RT-PCR positive. 

2. As per section 3.4, there was data on igG positivity and IgM negativity for 10 samples from another commercial kit. What about others (totally 42 SARS CoV2 positive used).

The patient samples were acquired based on RT-PCR result thus not all samples were tested for antibodies.

3. The study was designed to test for SARS CoV2 IgM, but Table 1 shows data for IgG. What about IgM?

Sorry, it was an error. The table was corrected. The results shown are IgM tests instead of IgG. Sorry for the confusion.

---

## [Decision Letter · Decision Letter 1]

19 Jul 2021

A High-throughput Microsphere-based Immunoassay of Anti-SARS-CoV-2 IgM Testing for COVID-19 Diagnostics

PONE-D-21-09355R1

Dear Dr. Sha,

We’re pleased to inform you that your manuscript has been judged scientifically suitable for publication and will be formally accepted for publication once it meets all outstanding technical requirements.

Kind regards,

Chandrabose Selvaraj, Ph.D.

Academic Editor

PLOS ONE

Additional Editor Comments (optional):

Reviewers' comments:

Reviewer's Responses to Questions

**Comments to the Author**

1. If the authors have adequately addressed your comments raised in a previous round of review and you feel that this manuscript is now acceptable for publication, you may indicate that here to bypass the “Comments to the Author” section, enter your conflict of interest statement in the “Confidential to Editor” section, and submit your "Accept" recommendation.

Reviewer #1: All comments have been addressed

Reviewer #2: All comments have been addressed

Reviewer #3: All comments have been addressed

2. Is the manuscript technically sound, and do the data support the conclusions?

Reviewer #1: (No Response)

Reviewer #2: Yes

Reviewer #3: Yes

3. Has the statistical analysis been performed appropriately and rigorously? 

Reviewer #1: (No Response)

Reviewer #2: Yes

Reviewer #3: N/A

4. Have the authors made all data underlying the findings in their manuscript fully available?

Reviewer #1: (No Response)

Reviewer #2: Yes

Reviewer #3: Yes

5. Is the manuscript presented in an intelligible fashion and written in standard English?

Reviewer #1: (No Response)

Reviewer #2: Yes

Reviewer #3: Yes

6. Review Comments to the Author

Reviewer #1: (No Response)

Reviewer #2: (No Response)

Reviewer #3: Revision has answered all the reviewer questions and provided clarifications where needed. There are no outstanding issues to be looked onto.

---

## [Editor Report · Acceptance letter]

24 Aug 2021

PONE-D-21-09355R1 

A High-throughput Microsphere-based Immunoassay of Anti-SARS-CoV-2 IgM Testing for COVID-19 Diagnostics 

Dear Dr. Sha:

I'm pleased to inform you that your manuscript has been deemed suitable for publication in PLOS ONE. Congratulations! Your manuscript is now with our production department. 

Kind regards, 

on behalf of

Dr. Chandrabose Selvaraj 

Academic Editor

PLOS ONE